

# Open access to regional geoid models: the International Service for the Geoid

Mirko Reguzzoni[1], Daniela Carrion[1], Carlo Iapige De Gaetani[1], Alberta Albertella[1], Lorenzo Rossi[1], Giovanna Sona[1], Khulan Batsukh[1], Juan Fernando Toro Herrera[1], Kirsten Elger[2], Riccardo Barzaghi[1], Fernando Sansó[1]

[1]International Service for the Geoid, Department of Civil and Environmental Engineering, Politecnico di Milano, Milan, 20133, Italy

[2]GFZ German Research Centre for Geosciences, Telegrafenberg, 14473, Potsdam, Germany

*Correspondence to*: isg@polimi.it

**Abstract.** The International Service for the Geoid (ISG, https://www.isgeoid.polimi.it/) provides free access to a dedicated and comprehensive repository of geoid models through its website. In the archive, both the latest releases of the most important and well-known geoid models, as well as less recent or less known ones, are freely available, giving to the users a wide range of possible applications to perform analyses on the evolution of the geoid computation research field. ISG is an official service of the International Association of Geodesy (IAG), under the umbrella of the International Gravity Field Service (IGFS). Its main tasks are collecting, analysing and redistributing local, regional and continental geoid models and providing technical support to people involved in geoid-related topics for both educational and research purposes. In the framework of its activities, ISG performs research taking advantage of its archive and organizes seminars and specific training courses on geoid determination, supporting students and researchers in geodesy as well as distributing training material on the use of the most common algorithms for geoid estimation. This paper aims at describing the data and services, including the newly implemented DOI Service for geoid models (https://dataservices.gfz-potsdam.de/portal/?fq=subject:isg), and showing the added value of the ISG archive of geoid models for the scientific community and technicians, like engineers and surveyors (https://www.isgeoid.polimi.it/Geoid/reg_list.html).

## 1 Introduction

The geoid, an equipotential surface of the Earth gravity field (Heiskanen and Moritz, 1967, Sansò and Sideris, 2013) has important applications in engineering for the definition of physical heights (Sansò et al., 2019), for example to compute them from ellipsoidal heights observed by GNSS techniques, and in geosciences, for example to determine the ocean geostrophic currents (Bingham et al., 2008, Knudsen et al., 2011). Because of its definition, the geoid is naturally modelled as a surface at a global level, typically through a truncated spherical harmonic expansion. The use of base functions on a spherical domain is justified by the fact that the geoid undulation is an anomalous quantity with respect to a given reference ellipsoid, while the



truncation of the series expansion depends on the spatial resolution of the model, e.g. a maximum spherical harmonic degree of 2160 means an angular resolution of about 5 arc-minutes (Pavlis et al., 2012). The coefficients of this spherical harmonic expansion, representing a global geopotential model, have been estimated since the mid of the last century, especially

exploiting satellite techniques (Merson and King-Hele, 1958, Barzaghi et al, 2015a), such as satellite orbit tracking and radar altimetry. With the advent of the new century, dedicated satellite gravity missions, i.e. CHAMP, GRACE and GOCE, were launched, significantly improving the knowledge of the gravity field. In this framework, the need of open accessing global models estimated by different research groups with different techniques led to the establishment of the International Centre for Global Earth Models (ICGEM) (Ince et al., 2019), an official IAG service with the aim of distributing global models and

also providing computational and visualization online tools.

However, global models from dedicated satellite missions are limited in terms of spatial resolution due to the dampening of the gravity signal with the orbital height (Pail et al., 2010, Pail et al., 2011). The use of ground gravity data allows to increase the global model spatial resolution, but, as a consequence of the inhomogeneous spatial distribution of these data (Pavlis et al., 2012), a certain level of regularization to compute the coefficients of global base functions is required. This regularization can

affect the quality of the geoid estimation also in areas where data spatial density is higher. For these reasons, and also for historical reasons related to the way in which the heights and the gravity data were observed, each country usually has its own geoid model. Biases are typically present between geoids of neighbouring countries, accounting for the different used conventions, e.g. the different reference tide gauge (Rummel and Teunissen, 1988, Sansò and Usai, 1995, Barzaghi et al., 2015b). These regional/local geoids can describe high frequency features that cannot be described by global models without

introducing an ultra-high maximum spherical harmonic degree and, consequently, a strong numerical regularization, because this high-resolution information is not homogeneously available at a global scale.

Regional geoid models are typically given as point-wise values over geographical grids or over sparse points (Forsberg and Tscherning, 2008). Like in the case of global models, there was a strong need from the scientific community, but also from the civil society for engineering applications, to access regional geoid models. In the framework of IAG, the International Service

for the Geoid (ISG) has the task of providing an open access to such information through the establishment and the maintenance of a geoid repository.

Regional geoid models can be mainly used for height conversion from ellipsoidal heights to orthometric or normal heights, or vice versa (Sansò et al., 2019). They can be further used for cross-check validation among them, for comparison with global models and, in general, for providing solutions that can be different from the official ones to the scientific community. To

reach these objectives, ISG developed a specific website with a geographic database.

The geoids archived by ISG correspond to Countries or Regions and data can be accessed via a list or by selecting them on a map. The repository is under the process of being implemented through a database in the framework of a WebGIS. To provide interoperable and reusable models, ISG is focussing on the provision of clear and complete metadata (Chan and Zeng, 2006, Longhorn, 2005) and the use of open, standardized and self-explanatory data formats (Cerri and Fuggetta, 2007).



The purpose of this paper is to introduce the reader to the service, by quickly revising the main steps of its history and of its activities performed inside the geodetic community (Section 2). Section 3 focuses on the ISG services: (1) the geoid repository, explaining its structure and current status, the data formats, with the description of the provided metadata, (2) the database indexing and DOI service, (3) the available height conversion online service and (4) the schools on the geoid determination. Finally, some examples of possible assessments exploiting the available models, implementing comparisons, and computing

statistics are provided in Section 4. The paper closes with some general considerations and future perspectives of the service.

## 2 ISG, a service of the International Association of Geodesy

The International Service for the Geoid (ISG) is an official service of the International Association of Geodesy (IAG). It was founded in 1992 as IGeS (International Geoid Service) with the aim to be one of the operative arms of the International Geoid Commission (IGeC). The service is provided by a main centre at Politecnico di Milano and by individual scientists. ISG

activities are in the framework of the International Gravity Field Service (IGFS), which includes other research centres and services: BGI (Bureau Gravimetrique International, France), IGETS (International Geodynamics and Earth Tide Service, France), ICGEM (International Centre for Global Earth Models, Germany), COST-G (International Combination Service for Time-variable Gravity Fields, Switzerland), IDEMS (International Digital Elevation Model Service, USA), IGFS-TSC (Technical Support Centre of IGFS, USA). IGFS is a unified IAG service aiming at collecting, validating and distributing data

and software for the purpose of determining, with various degrees of accuracy and resolution, the gravity potential of the Earth, or any of its functionals, and the surface of the Earth.

The main tasks and activities of ISG are:

- to collect geoid estimates worldwide, validate them when possible, and disseminate them to users. Other auxiliary data, like global gravity models, useful for the geoid determination, may also be collected by ISG, without

redistributing them if they are already provided by other IAG services. Since summer 2020, ISG is offering to assign digital object identifiers (DOI) to their geoid models. This DOI Service has been establish in collaboration with GFZ Data Services and allows geoid model being cited in publications (e.g. Barzaghi et. al, 2020a and b);

- to collect, test and, when allowed, distribute software for the geoid determination;

- to conduct research on methods for the geoid determination, also defining optimal procedures for merging all available

data and models;

- to organize international schools on geoid determination addressing both theoretical and practical topics, possibly every two years. During the schools, students are trained in the use of the relevant software for geoid computation;

- to support agencies or scientists in computing local and regional geoid models, especially in developing countries, also organizing special training courses;

- to disseminate training material and software on geoid computation, e.g. lecture notes of the schools;



- to issue the Newton's Bulletin, which has a technical and applied nature, collecting papers and reports on gravity and geoid;

- to establish and update a website to present the service activities, show and distribute the geoid models, software and publications, announce news and the organization of international schools on geoid determination; the geoid model
distribution is also carried out via the catalogue of GFZ Data Services for DOI-referenced models only;

- to provide users with on-line services through the website, like the height conversion service exploiting any of the publicly available geoid models in the geoid repository (see Sec. 3.2).

## 3 ISG Services

The main services offered by ISG are the geoid repository, the database indexing and DOI service, the height conversion online
service and the schools on the geoid determination. The formers are data-oriented, while the latter is for educational purposes.

### 3.1 The ISG geoid repository

ISG manages and preserves an openly accessible repository of regional, national and continental geoid models at a worldwide scale. The repository aims at storing and redistributing geoid models in standardised data format, providing also ancillary information useful for gravity related analysis. Most models can be freely downloaded, some of them require the author's
permission to be accessible and few are private and cannot be distributed. Consequently, they are classified as public, on-demand or private, respectively. The ISG geoid repository currently[1] stores 226 geoid models (158 gravimetric models, 8 geometric models and 60 hybrid models) under different policies (168 public, 21 on-demand, and 37 private). Monthly, the webpages are visited about 1200 times. The coverage area is almost worldwide, with resolution grids up to 0.5 arc-minutes as shown in Fig. 1 (in log10 scale).

Particular attention is devoted to metadata and data interoperability. When stored, the geoids are collected both in the format provided by the owners and in ISG format, a standardized ASCII format developed on purpose. Details on this format will be described in the next subsection. Moreover, for each geoid, metadata are collected and archived, like the names of the authors, the publication year of the model, key reference publication(s) and a brief description on the computational method of the model. These pieces of information are then published on the website through the model-related web pages. An example of a
model landing page is shown in Fig. 2.

---

[1] These statistics are based on the repository status at 10th November 2020





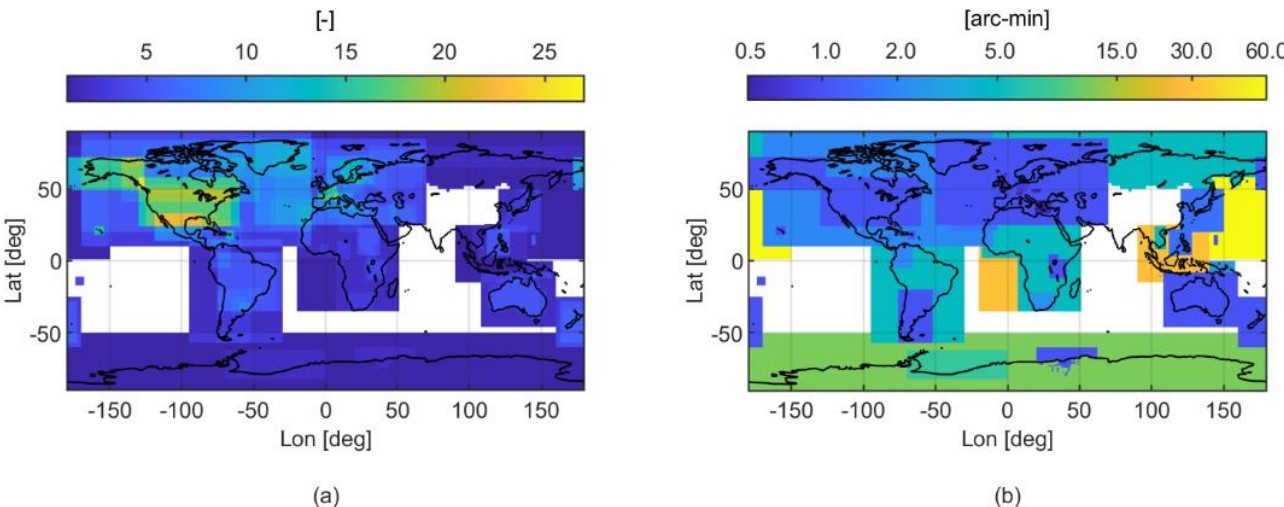

**Figure 1. Geoid repository. (a) number of available geoid models covering the same area, blank means missing data; (b) spatial resolution of the grid geoid models available at ISG. Colour bars show the highest spatial resolution available per location, log10 scale, units in arc-minutes;**




**Figure 2. Geoid repository, example of a webpage describing the content of the selected available geoid.**






The importance and the scientific relevance of such service does not only depend on the amount of available data but also on the completeness and clarity of the provided information. To this purpose, ISG provides geoid models in a format that has been designed to be easily understood and managed by the final users.

### 3.1.1 The ISG data format

The ISG format provides local/regional geoid/quasi-geoid models given as undulations with respect to a reference ellipsoid on a grid or sparse points. The file is in ASCII format with the extension .isg. The first version of the ISG ASCII format was released in 2015 and updated in 2018 (version 1.01). In July 2020, a major new release, version 2.0, was published, mainly introducing more metadata to better characterize the content of the file and also allowing to store sparse point data. All new models will be published with version 2.0. In addition, before an already published model is assigned with a DOI, it will be

transformed in ISG ASCII version 2.0

Each individual data file consists of three sections: a) the optional comment section, which starts at the beginning of the file and ends with the keyword "begin_of_head" (as a separator between the comment and the header sections); b) the header section, which contains textual and numerical parameters. The beginning of the header is marked by the keyword "begin_of_head", while the end of the header is marked by the keyword "end_of_head"; c) the data section including the

undulation values. To increase data interoperability, section (a) and (b) were designed with the same scheme of the .gfc file, distributed by ICGEM and providing global model coefficients (Ince et al., 2019).

In the comment section (a), three paragraphs are strongly recommended, the first one with the licence under which the data are distributed, the second one with the reference to cite when using the data, the third one indicating the data provider and the institution distributing the model.

In ISG format, the header section (b) is composed by structured metadata. It can be conceptually divided into three parts. The first contains textual metadata that are required to characterize the model, such as:

- the name of the model and the year of computation,
- the type of the model (gravimetric, geometric or hybrid),
- the classification between geoid and quasi-geoid,

- the fact that the data are sparse or gridded, and in case the ordering of the gridded data,
- the reference ellipsoid and datum, the reference frame, and the tidal system,
- the fact that the coordinates are geodetic or projected and in case the type of projection,
- the units of the undulation data and the coordinate units.

The second part contains numerical metadata that are mainly required to georeferencing the undulation values, such as

- the bounding box of the undulation dataset, i.e. minimum and maximum coordinates,
- the grid step and the number of rows and columns if the data are gridded (the number of rows can be used in sparse data to specify the number of points),
- the no-data value for missing points inside the grid structure.



Finally, the header contains information about the file, such as the and the format version. Metadata and their keywords depend
on the format version. The file format specifications for all the possible versions are available at a dedicated page on the ISG
website (http://www.isgeoid.polimi.it/Geoid/format_specs.html).

The data section was originally developed to contain the gridded undulation values, but from the format version 2.0 it is also
possible to store sparse data by providing the point coordinates along with the undulation values. In case of gridded data, the
point coordinates are defined in the header section and the undulation values are always stored row by row, being the default
ordering from North to South, each row going from West to East.

## 3.2 The ISG database indexing and DOI service

The sharing of reliable, citable and well-described research data are key elements for Open Science. The European Union is
raising the attention to the importance of data sharing and metadata through actions, policies and directives, such as INSPIRE
or the "European legislation on open data and the re-use of public sector information" (EU Open Data Directive). In addition,
the new Horizon Europe programme, which is about to start, is requiring not only the open access to research papers but also
the open access to the data. To effectively archive, discover and access data it is crucial to prepare and store complete metadata
and to have access to well-known archives with a stable link. In addition, to grant proper credit to research authors, it is very
important to allow for the data citation as well. Assigning Digital Object Identifiers (DOI) to research data is an important
instrument for this and best practice for FAIR Sharing Data (e.g. Fenner et al., 2019, Hodson et al., 2018, Wilkinson et al.,
2016). Datasets with assigned DOIs are fully citable in scholarly literature that enables tracking of data usage via citation
metrics and provides credit for researchers and institutions

For these reasons, ISG has established agreements with (1) Clarivate's the Data Citation Index™
(https://clarivate.com/webofsciencegroup/solutions/webofscience-data-citation-index/) and (2) with GFZ Data Services
(https://dataservices.gfz-potsdam.de/portal/), the research data repository for the Geosciences domain, hosted at GFZ German
Research Centre for Geosciences (https://www.gfz-potsdam.de/en/home/).

### 3.2.1 Data Citation Index

Clarivate's "Data Citation Index™" serves as "single point of access to quality research data from global repositories across
disciplines" that are "linked to literature articles in the Web of Science.™"
(https://clarivate.com/webofsciencegroup/solutions/webofscience-data-citation-index/). Geoid models of ISG are indexed in
the Data Citation Index and the accession number is indicated on the geoid model website (see lower left of Figure 3).

### 3.2.2 DOI Service

Since summer 2020, ISG, in collaboration with GFZ Data Services, has extended their services by offering the assignment of
DataCite DOIs to geoid models archived in the ISG geoid repository. This includes the generation and provision of
standardised, machine-readable metadata following international standards (DataCite, ISO19115) that are complementary to



the disciplinary metadata already collected for ISG geoid models and openly accessible via a standard application programming
interface (API). An individual DOI is assigned to each geoid (or quasi-geoid) model by GFZ Data Services, who is also serving
as an additional archive of the ISG repository. The agreed licence for geoid models is the Creative Commons Attribution 4.0
(CC BY 4.0) Licence. Before the DOI assignment, the model data are converted to the ISG 2.0 format, which includes the
citation and licence information in the header of the data. Geoid models assigned with DOI can be additionally discovered via

the catalogue of GFZ Data Services and via the repository machine-readable DOI landing pages that are embedding schema.org
(https://schema.org/). To this aim, GFZ Data Services has created a new internal "datacentre" for ISG geoid models
(https://dataservices.gfz-potsdam.de/portal/). The DOI landing pages at GFZ Data Services are closely cross-linked with the
model pages at ISG, as well as with key articles or reports describing the geoid models. Different type of models, e.g. geoids
and quasi-geoids, gravimetric and hybrid solutions, computed in the same framework are cross-referenced on the DOI landing

pages and in the DataCite metadata of each model. The DOI links are also added to the model pages at ISG (see figure 3)

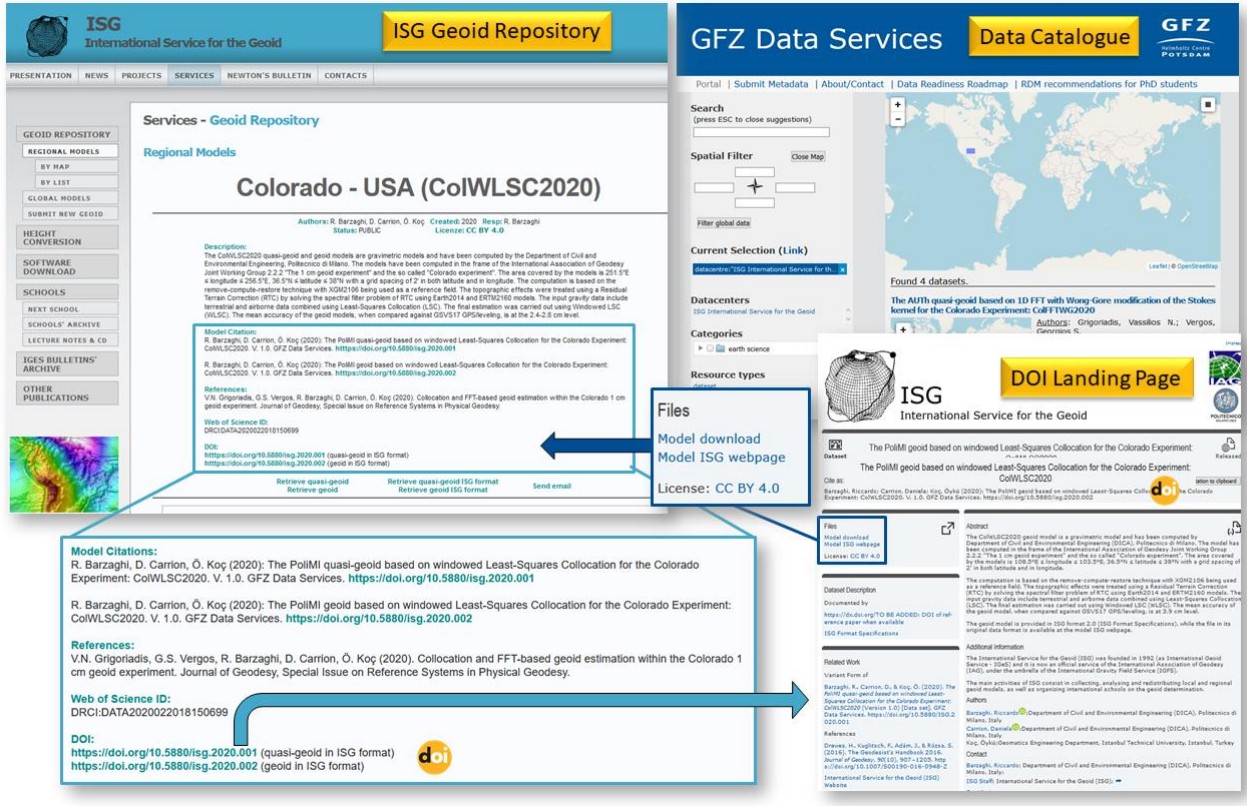

**Figure 3: Overview on the relation between ISG and GFZ Data Services for DOI-referenced geoid models of the
Colorado experiment (ColWLSC2020). This project includes a quasi-geoid model (Barzaghi et al., 2020a) and a geoid
model (Barzaghi et al., 2020b). Both models can be accessed via the dedicated webpage in the ISG geoid repository (on**

**the left) and via the DOI landing pages in the GFZ Data Services (on the right). The "File" section of the DOI landing**



**page includes the links to the model file and the corresponding webpage in the ISG repository. On the other side, the ISG model webpage is enhanced with the recommended citations of the DOI-assigned models and the links to the DOI landing pages at GFZ Data Services.**


### 3.3 The ISG height conversion web-service

ISG offers a height conversion web-service to the users. They can provide the coordinates of one or more points (in the latter case through a CSV file containing three columns, namely latitude, longitude and height to be converted) and, after selecting the geoid model and the interpolation method, the web-service returns the conversion from ellipsoidal to orthometric height or

vice versa. Once the user provides the point coordinates, only the geoid models containing at least one of these points are listed and can be selected by the user for the height conversion. This is possible by exploiting the model bounding box information that is available in the model file header as defined according to the ISG format.

As for the algorithmic point of view, the conversion is based on the formula $H = h - N$, relating the ellipsoidal height $h$ and the orthometric height $H$ through the geoid undulation $N$. The interpolation method can be chosen by the user between a

bilinear interpolation among the four model points that are the closest to the input one, or an inverse distance weighted interpolation.

As for the software implementation point of view, the web-service is divided into front-end and back-end, the former providing a user interface and the latter performing the calculations. The front-end is the "visible" part of the application, it is implemented by using an HTML page and JavaScript. The HTML page contains a form with all the needed fields for the height

conversion according to the web-service created on the back-end (see Figure 4). The interface is designed to change as the user interacts with the application and selects the different options (single or multiple point coordinates). There are also checks on the input file size and format when the user asks for the conversion of more than one point.



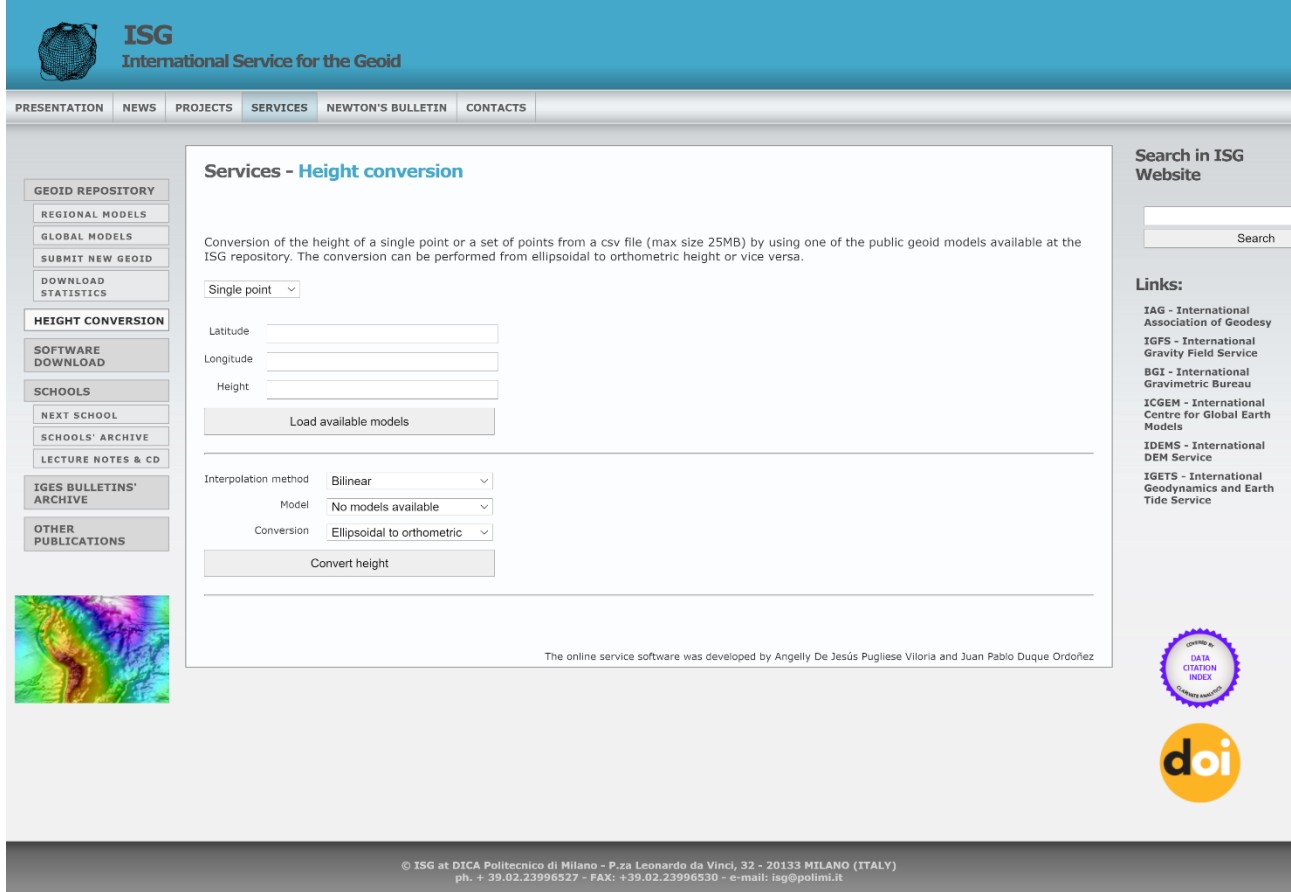

**Figure 4. Web page to access the ISG height conversion web-service**

The back-end is the core of the web-service, performing the required computation without increasing the burden of the front-end. In this way the web-service can be modified or updated without interfering with the front-end. In order to implement the back-end, a REST API (Representational State Transfer Application Programming Interface) was created in Django, which allows us to perform mathematical calculations using Python with the NumPy library. Four different endpoints were created for the geoid model research and the height conversion, both for a single point and a set of points. A sketch of the logical structure of this web-service is provided in Figure 5.

All requests from the front-end to the back-end rely on HTTP POST method, i.e. enclosing the data in the body of the request messages instead of storing it, while the answers from the back-end are transmitted through a JSON file which content is directly visualized in the HTML page.





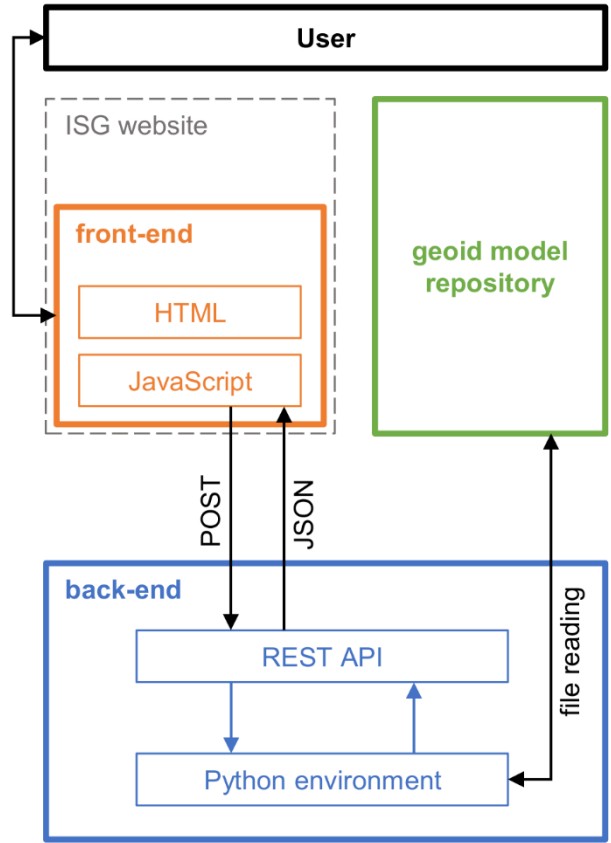

**Figure 5. Logical structure of the height conversion web service**

**3.4 Geoid schools**

Geoid schools, i.e. the organisation and support of technical schools on geoid estimation and related topics, are the earliest and central educational task of ISG. Beginning with the first International School on "The determination and use of the geoid" (1994 in Milan), ISG's Geoid Schools have developed to an international reference for geoid computation training and gravity field modelling. The general purpose of the full-week intensive geoid school is to prepare graduate students, young researchers, employees of national agencies and services or industry staff, to the computation and usage of gravimetric geoid models for

scientific and technical applications in geodesy. The schools provide a good opportunity to familiarize with the latest developments in geoid determination and to improve international contacts and collaborations among scientists dealing with gravity field modelling. Theoretical lectures are followed by computer exercises using software that is also distributed by ISG. The general frame of the lectures is the following:

- introduction to physical geodesy;

- computation and use of high degree and ultra-high degree geopotential models;
- geoid computation using Stokes' integral and collocation;



- • terrain effects in geoid estimation;

- • Fast Fourier Transform (FFT) techniques in geodesy;

- • seminars on specific topics depending on the research fields of the guest tutors.

Lecture notes of the courses are distributed to the participants, as well as the required software and the exercises. Such material can be also provided upon request through the ISG website. Since 1994, 12 editions of the geoid school have been organized, held in 10 countries and attended by 340 people (see Table 1):

**Table 1. List of the Geoid Schools organized by ISG since its foundation.**

| Location | Date | Attendees |
|---|---|---|
| Mongolian University of Science and Technology, Ulaanbaatar (MNG) | 6-10/06, 2016 | 30 |
| Universidad Técnica Particular de Loja, Loja (ECU) | 7-11/10, 2013 | 15 |
| Research Institute Elektropribor, S. Petersburg (RUS) | 28/06 – 2/07, 2010 | 15 |
| National University of La Plata, La Plata (ARG) | 7-11/09, 2009 | 23 |
| Politecnico di Milano, Como Campus, Como (ITA) | 15-19/09, 2008 | 25 |
| Niels Bohr Institute, University of Copenhagen, Copenhagen (DNK) | 19-23/06, 2006 | 24 |
| Budapest University of Technology and Economics, Budapest (HUN) | 31/01 – 5/02, 2005 | 49 |
| University of Thessaloniki, Thessaloniki (GRC) | 30/08 – 5/09, 2002 | 30 |
| Department of Survey and Mapping Malaysia, Johor - Bahru (MYS) | 21-25/02, 2000 | 41 |
| Politecnico di Milano, Milan (ITA) | 15-19/02, 1999 | 23 |
| Instituto Brasileiro de Geografia e Estatística, Rio de Janeiro (BRA) | 10-16/09, 1997 | 31 |
| Politecnico di Milano, Milan (ITA) | 10-14/10, 1994 | 34 |


Beyond the international geoid schools, also ad-hoc training periods are carried out by ISG in order to support foreign institutions and scientists in the field of geoid computation.

**4 Geoid comparative analyses based on the ISG repository**

Relative comparisons can be carried out between the available geoids stored in the ISG repository. Such comparisons can
reveal useful information, like possible mismatches between estimated geoids, for example when two models referring to the same area are computed at different epochs or with different techniques or published by different authors. Additionally, local or regional models can be compared with global gravitational models. In the following, examples of such kind of relative assessments are described.

**4.1 Assessment of the evolution over time of the same geoid computed by the same authors.**

As an example of evolution over time, the Japanese geoid can be considered. The ISG geoids repository collected several versions of Japanese national geoids computed by the Geographical Survey Institute. Among them, the group of hybrid geoids published in 1996, 2000 and 2011 can be compared to evaluate their evolution over time. The first geoid is GSIGEO96 (Fukuda et al., 1997). It is provided on a 3'x3' grid and it is referred to the GRS80 ellipsoid in the ITRF89 frame (Fig. 6a). It has been

computed by fitting the JGEOID93 gravimetric geoid model to the nationwide network of 806 sites via Least Squares

Collocation (LSC). The second geoid, named GSIGEO2000 (Kuroishi, 2000, Fig. 6b), is provided on a 1'x1.5' grid and it is

referred to the GRS80 ellipsoid in the ITRF94 frame at epoch 1997.0 (Fig. 6b), i.e. the Japanese Geodetic Datum 2000

(JGD2000). It has been computed by fitting the JGEOID2000 gravimetric geoid model to the nationwide net of GPS/levelling

data via LSC. The third geoid is the GSIGEO2011 (Miyahara et al., 2014). Again, this model is provided on a 1'x1.5' grid and

it is referred to the GRS80 ellipsoid in the ITRF94 frame at epoch 1997.0 (Fig, 6c), i.e. the Japanese Geodetic Datum 2000

(JGD2000). In this case, it has been computed by fitting the JGEOID2008 gravimetric geoid model to GNSS/levelling data at

971 sites (786 GEONET stations, 156 benchmarks and 29 tidal stations) via LSC.

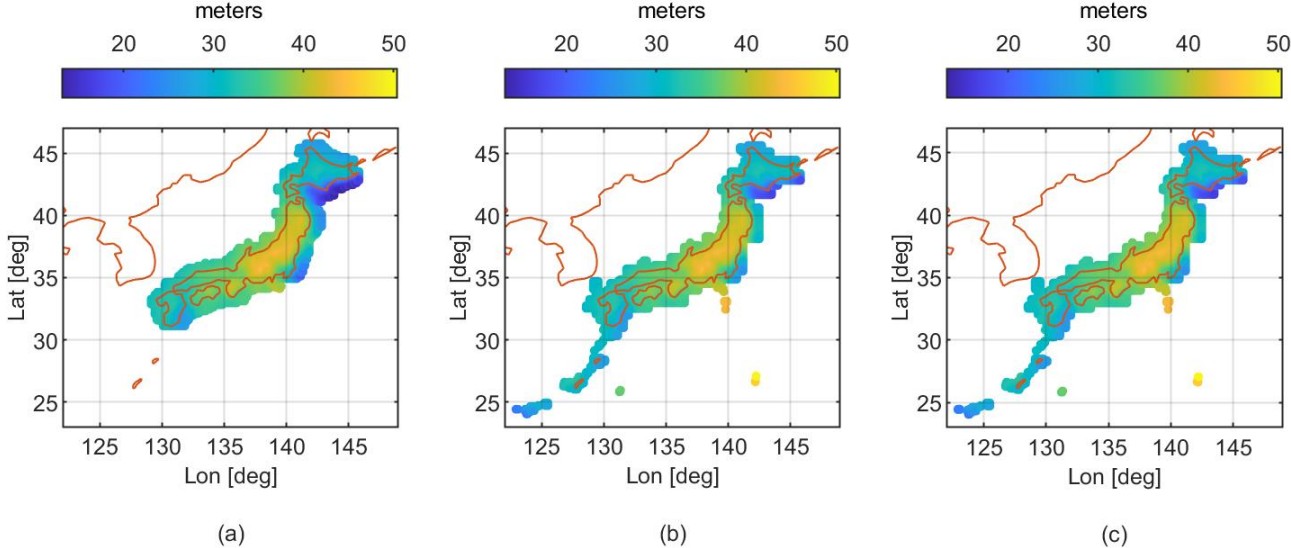

**Figure 6. Evolution of Japanese geoid models over time. The three panels represent the undulation with respect to the reference ellipsoid of: (a) GSIGEO96, (b) GSIGEO2000 and (c) GSIGEO2011**


Due to the different resolutions and grid boundaries, the comparison has been done stepwise by interpolating the most recent

geoid on the same grid of the previous model. Once the grids have been made consistent, a plane fitting their differences has

been estimated and removed before statistics computation. This was done for reducing the impact of possible mismatches due

to different reference frames and height datums. Table 2 shows the main statistics of the differences between two consecutive

versions of the same geoid model.

**Table 2. Statistics of the differences between the analysed Japanese geoids.**

| Geoid A | Geoid B | Grid Resolution | Std [m] | Min [m] | Max [m] |
|---------|---------|-----------------|---------|---------|---------|
| GSIGEO96 | GSIGEO2000 | 3' x 3' | 0.184 | -1.017 | 1.060 |
| GSIGEO2000 | GSIGEO2011 | 1' x 1.5' | 0.104 | -0.610 | 0.891 |



From these statistics, it can be noted that the two refinements of the GSIGEO geoid model have a different impact. The standard
deviation of the residuals between the "editions" 1996 and 2000 is almost 19 cm while between the "editions" 2000 and 2011
is halved, about 10 cm. This is mainly due to a better quality and larger terrestrial gravity database and also to an increase of
the computational power that allowed to better estimate high frequency behaviours around the consolidated estimates of the
long wavelength. It is interesting to look at the spatial distribution of the residuals, shown in Fig. 7. Referring to GSIGEO96-

00, high amplitude residuals are concentrated along the eastern border, with ranging down to about -1.0 meter; referring to
GSIGEO00-11, residuals are generally smaller but still localized in the eastern and northern borders, with opposite amplitude
with respect to GSIGEO96-00; areas where residuals of GSIGEO96-00 and GSIGEO00-11 are different for more than 50 cm,
suggest disagreement between the two refinements.

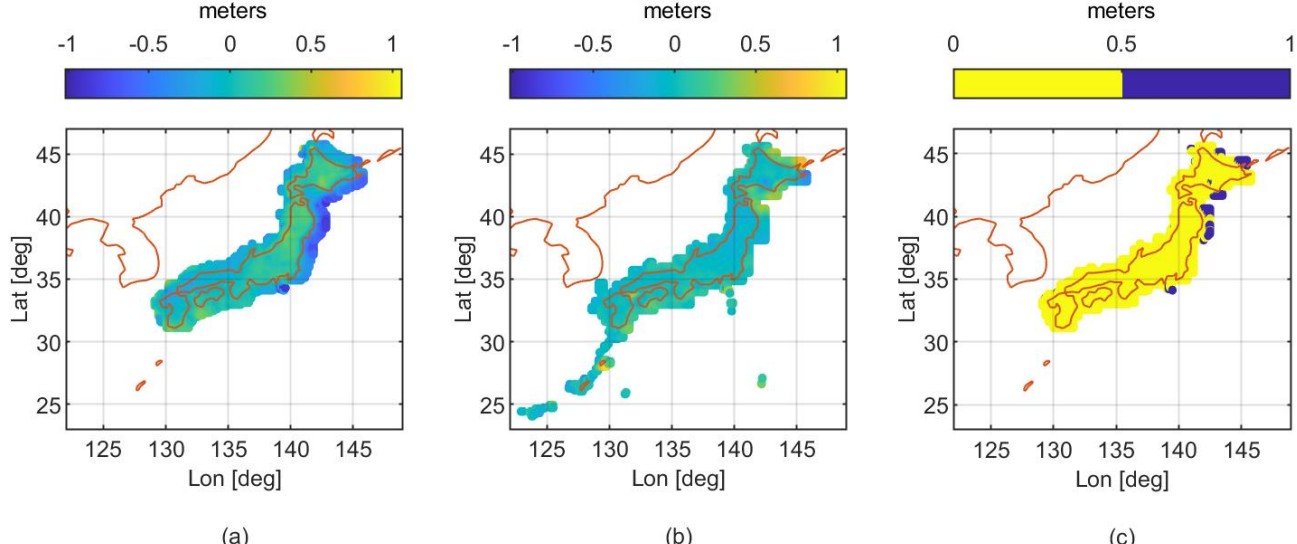

**Figure 7. Difference between the analysed Japanese geoids. (a) GSIGEO96 – GSIGEO2000; (b) GSIGEO2000 – GSIGEO2011; (c) Combined agreement between the considered Japanese geoids. In yellow areas the difference between GSIGEO96–00 and GSIGEO00-11 is less than 50 cm while in blue areas is more than 50 cm.**

### 4.2 Comparison of geoids computed by different authors and with different techniques

The ISG repository contains also collections of the same national geoid computed with different techniques by different
authors. As an example, one can consider the three geoids of the Sudan and South-Sudan publicly available in the ISG database.
In this case, it can be interesting to assess the agreement between them and evaluate possible mismatches and inconsistencies.
Sudan KTH-SDG08 (Abdalla, 2009) is the first Sudanese gravimetric geoid model considered in this kind of comparison. It
has been computed by applying a least squares modification of Stokes formula. Two sets of in-situ regional gravity data have
been used: the first one has been provided by Geophysical Exploration Technology (GETECH) group from the University of



Leeds, UK. This dataset contains gravity observations that do not cover uniformly the Sudanese region. The second dataset is provided by the International Gravimetric Bureau (BGI) and contains only few scattered points measured over the neighbouring countries. The long-wavelength contribution has been modelled by using two different geopotential models: The EIGEN-GRACE02S satellite-only model (Reigber et al., 2005) has been used in the modified Stokes formula, whereas the EIGEN-GL04C combined model (Foerste et al., 2006) has been used to enrich the local gravity coverage over the areas with

missing data. SRTM has been used to compute the topography effects on the geoid. The KTH-SDG08 model is provided on a 12'x12' grid over the computation area bounded by 4° N to 23° N in latitude and 22° E to 38° E in longitude. The geoid heights are given with respect to the GRS80 reference ellipsoid (Fig. 8a).

The second Sudanese geoid, called simply Sudan has been computed by Fashir and Kadir in 1998 (Fashir and Kadir, 2000). This geoid model has been computed for the whole region of Sudan and South-Sudan, ranging from 4° N to 24° N in latitude

and 22° E to 38° E in longitude. The EGM96 geopotential model truncated to degree and order 70 has been combined with surface gravity data and modified Stokes's kernel to generate the geoid file on a 10'x10' grid (Fig. 8b).

The third model for this comparison is Sudan SUD-GM2014 (Godah and Krynski, 2015). This has been computed on a 5'x5' grid using the GO_CONS_GCF_2_TIM_R4 global geopotential model (Pail et al., 2011), with terrestrial mean free-air gravity anomalies and the high-resolution SRTM30_PLUS global digital elevation model. The computations of the Sudan SUD-

GM2014 have been performed using the remove-compute-restore procedure and the LSC method (Fig. 8c).

As previously described, the comparison has been carried out interpolating the three geoids on the area covered by all of them and with a grid spacing equal to the lowest available resolution (12'x12'). Relative differences have been computed and analyses have been conducted on the residuals obtained after removing the interpolating plane, which models possible mismatches due to different datum and reference frame.

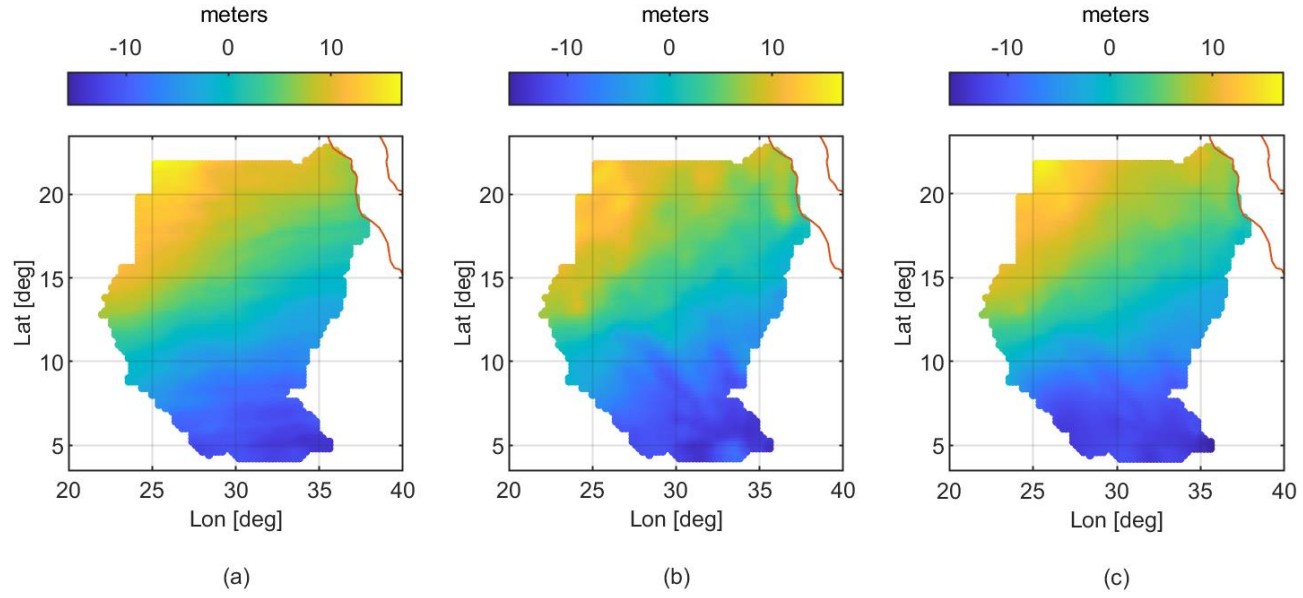




**Figure 8. Sudanese geoid models. The three panels represent the undulation with respect to the reference ellipsoid of (a) Sudan KTH-SDG08; (b) Sudan; (c) Sudan SUD-GM2014. All the models were considered only inside the national borders of Sudan and South-Sudan.**

Table 3 shows the statistics of the residuals obtained with the three different combinations. Their standard deviations are quite similar and less than 1 m. What is interesting to see is that, despite most of the grid points show residuals below this value (ranging from 75% to 82%), their spatial distribution is quite different, as shown in Figure 9.

**Table 3. Statistics of the difference between the analysed Sudanese geoids.**

| Case | Geoid A | Geoid B | Grid Resolution | Std [m] | Min [m] | Max [m] | Abs < 1m |
|------|---------|---------|-----------------|---------|---------|---------|----------|
| A | Sudan KTH-SDG08 | Sudan | 12' x 12' | 0.942 | -4.628 | 3.377 | 75% |
| B | Sudan KTH-SDG08 | Sudan SUD-GM2014 | 12' x 12' | 0.829 | -2.294 | 3.686 | 82% |
| C | Sudan | Sudan SUD-GM2014 | 12' x 12' | 1.001 | -3.797 | 4.180 | 77% |


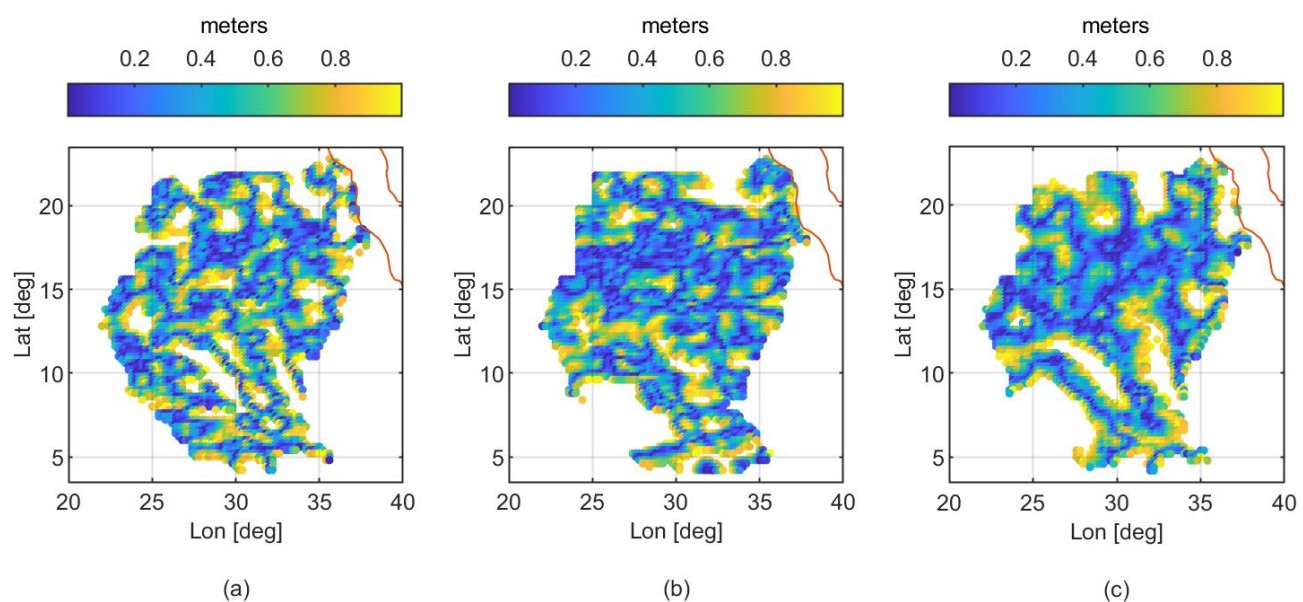

(a)                                          (b)                                          (c)

**Figure 9. Sudanese geoids comparison, residuals with absolute value less than 1 meter. (a) case A; (b) case B; (c) case C.**

It can be noted as Sudan KTH-SDG08 and Sudan SUD-GM2014 show a better agreement (case B) apart from the south-western area where differences are remarkable. This area shows disagreement also in case C (Fig 9c), thus implying that Sudan SUD-GM2014 model behaves significantly different from the others in this area. Along the north-eastern border of Sudan, residuals obtained in all the three comparisons are large, which might be attributed to a different gravity data coverage.

Such kind of analysis does not aim at assessing which is the best model but provide useful information to evaluate areas that
can be considered as more reliable than others when similarly modelled by different geoid models. Figure 10 shows the areas
where the residuals of the three comparisons are similar. In this analysis the three models are considered as similar if their
geoid values are different by less than the minimum standard deviation of the residuals of the three comparisons, here 0.829
m (case B). Under this assumption, 48% of the grid points have a good agreement among all three considered geoids. This
allows the conclusion that these areas are, more likely, the most reliable estimates of the geoid.

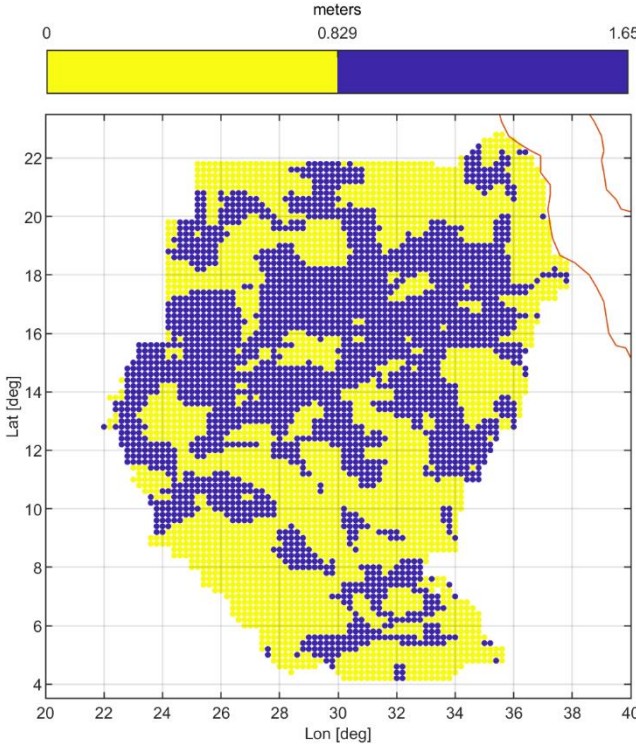


**Figure 10. Combined agreement between Sudanese geoids. Yellow grid points show where all three considered geoids
differ for a value below the minimum obtained standard deviation in the three trials, then considered in agreement.**

### 4.3 Comparison of geoids on overlapped areas

Another interesting comparison can be performed between geoid models at different scales (e.g. regional, national and local)
on overlapped areas. This can be useful to assess the agreement between them and detect possible problematic areas requiring
further investigations. Figure 11 shows three geoid models overlapped on the test area in South America where the assessment
will be carried out.



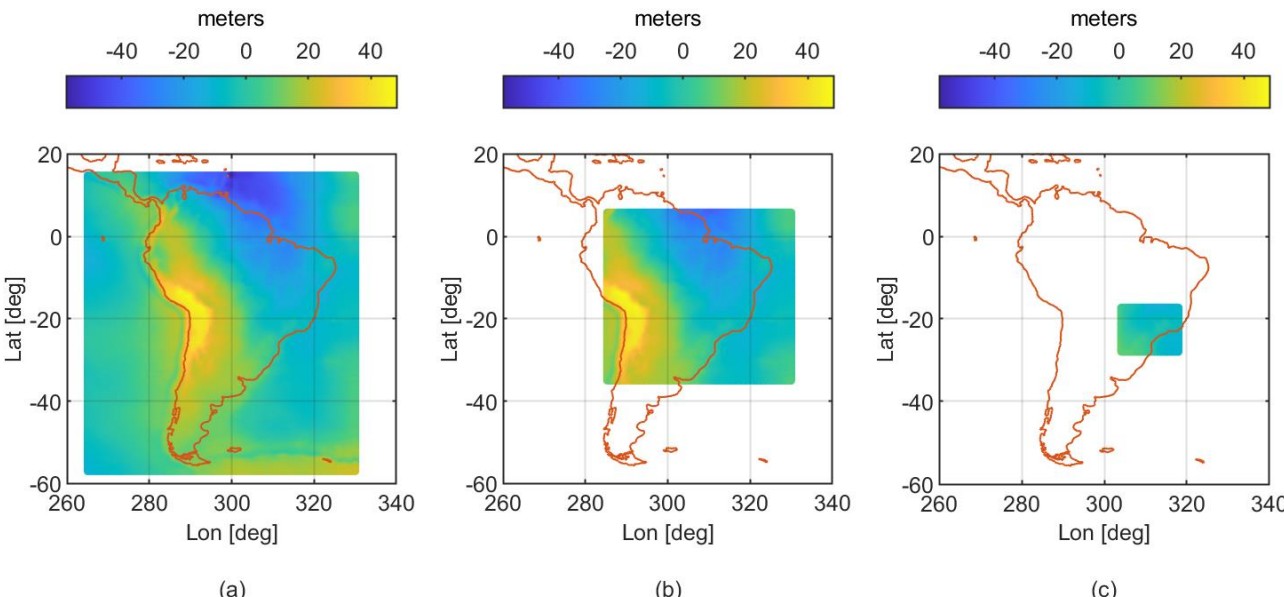

**Figure 11. Continental, national and local geoid models. The three panels represent the undulation with respect to the**
**reference ellipsoid of: (a) South America geoid model; (b) MAPGEO2015; (c) GEOID-SP.**

The first model is the continental South America geoid model (de Matos et al., 2014), computed on a 5'x5' grid by the remove-restore technique using 925,878 gravity data points (free-air gravity anomalies), EGM2008 (Pavlis et al., 2012) up to degree and order 150, the SAM3s_v2 DTM for the computation of terrain correction and other topographic and atmospheric effects.

The mean free-air gravity anomaly in a 5' grid over continent was derived from the complete Bouguer Anomalies, while free-air anomalies over the ocean obtained from satellite gravity model DNSC08 (Andersen et al., 2009). The short wavelength component was estimated by using FFT with the modified Stokes integral through spheroidal Molodenskii-Meissl kernel modification. Geoidal heights are referred to WGS84 ellipsoid.

Within the same area, the second model is a national model, the official Brazilian geoid called MAPGEO2015 (Blitzkow et

al., 2016). It has been computed as a cooperation between IBGE (Instituto Brasileiro de Geografia e Estatística) and EPUSP (Escola Politeécnica Universidade de São Paulo). It is a 5'x5' grid of geoidal undulations referred to WGS84 ellipsoid. The gravity dataset consists of 947,953 terrestrial gravity points (450,589 in Brazil), the terrain effect has been computed using a digital terrain model based on SRTM and the EIGEN-6C4 global geopotential model (Foerste et al., 2014) up to degree and order 200 has been used in modelling the low-frequency component of the gravity field. The remove-compute-restore

procedure was used and the short wavelength components were computed by Stokes integration via FFT technique.

Finally, the third model for this comparison is a local model, the GEOID-SP model (Guimarães et al., 2014) of the State of São Paulo, Brazil. It has been computed on a 5'x5' grid by the remove-restore technique using two different methodologies: the FFT with the modified Stokes integral through spheroidal Molodenskii-Meissl kernel modification and the LSC. In this

comparison, the one computed with the first methodology has been considered. The reference geopotential model is
GO_CONS_GCF_2_DIR_R5 (Bruinsma, S. L. et al, 2013) up to degree and order 200. The SAM3s_v2 DTM has been used
for the computation of terrain correction and other topographic and atmospheric effects. The mean free-air gravity anomaly on
a 5'x5' grid over the continent has been derived from the complete Bouguer Anomalies, while the free-air over the ocean were
obtained from satellite gravity model DTU10 (Andersen, 2010).

The comparison area is the one covered by the grid provided by the local GEOID-SP model and, since the others have been
computed on overlapped grids with the same resolution, no interpolation have been done to perform the comparison. Table 4
shows the main statistics of the residuals between the three models after removing the interpolating plane modelling possible
mismatches due to reference systems and height datum mismatches.

**Table 4. Statistics of the difference between the analysed geoids over the São Paulo region (BRA).**

| Geoid A | Geoid B | Grid Resolution | Std [m] | Min [m] | Max [m] |
|---------|---------|-----------------|---------|---------|---------|
| South America | Brazil | 5' x 5' | 0.078 | -0.574 | 0.723 |
| Brazil | Sao Paulo | 5' x 5' | 0.050 | -0.635 | 0.382 |
| South America | Sao Paulo | 5'x5' | 0.082 | -0.4967 | 0.412 |


As expected, the standard deviation of the residuals decreases by comparing more localized geoid models. In fact, the residuals
between the national and local geoid have standard deviation of 5 cm, while between the regional and the local one is about 8
cm. Figure 12 shows the spatial distribution of these residuals.

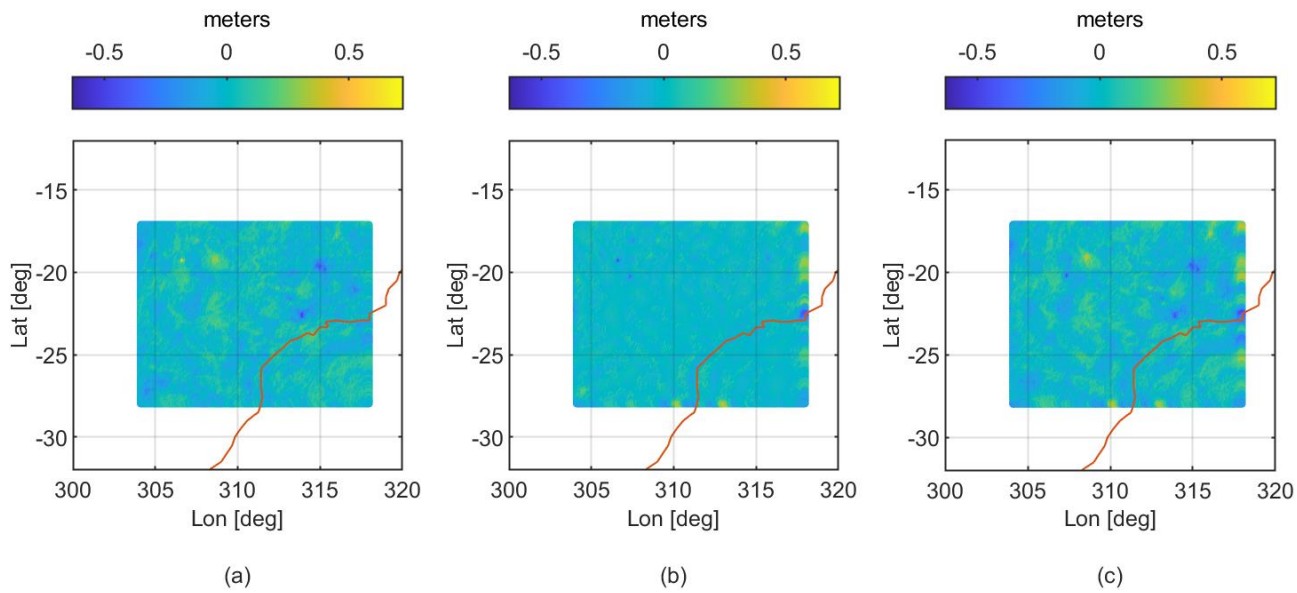

**Figure 12. São Paulo area geoids comparison, residuals between the considered geoids. (a) Continental - National; (b)
National - Local; (c) Continental - Local.**

Looking at the differences between the Brazilian and São Paulo state geoid models, edge effects are clearly visible and it can be noted that in the area around -19° in latitude and 307° in longitude there are spots where differences are particularly relevant.

This occurs also in the other combinations, leading to think that this is a quite problematic area for geoid estimation and should require further investigations by the scientific community.

### 4.4 Comparison of local geoids with global models

The agreement between local and global models can be assessed, as well. Global models can be used at their full resolution to synthesize the geoid at arbitrary points of a local area. This is an advantage with respect to local geoids but at cost of possible

mismodelling of local features. The comparison between local and global geoids at the same points gives the possibility of detecting those areas where the contribution of local geoids is more prominent. Figure 13 presents three geoid models taken as an example for such an assessment.

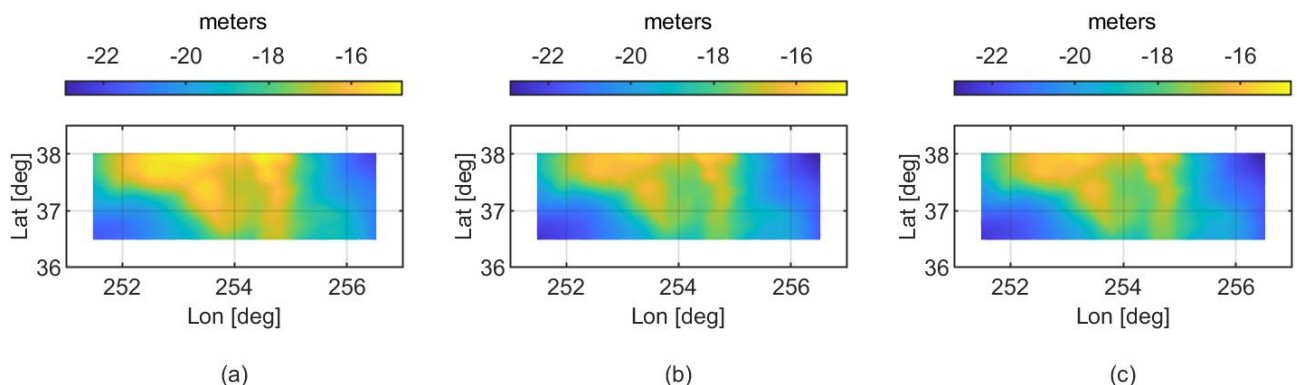

**Figure 13. Local and global geoid models. The three panels show the undulation over the Colorado (US) area with respect to the reference ellipsoid of: (a) ColWLSC2020; (b) EIGEN-6C4; (c) EGM2008.**

The first model is the recent ColWLSC2020 (Barzaghi et a., 2020b) gravimetric geoid model, computed by the Department of Civil and Environmental Engineering, Politecnico di Milano. This model has been computed in the frame of the International

Association of Geodesy Joint Working Group 2.2.2 "The 1 cm geoid experiment" and the so called "Colorado experiment". The area covered by the model is 251.5°E ≤ longitude ≤ 256.5°E, 36.5°N ≤ latitude ≤ 38°N with a grid spacing of 2' in both latitude and longitude. The computation is based on the remove-compute-restore technique with XGM2016 (Pail et al., 2017) being used as a reference field. The topographic effects were treated using a Residual Terrain Correction (RTC) by solving the spectral filter problem of RTC using the Earth2014 and ERTM2160 models. The input gravity data include terrestrial and

airborne data, that were combined by Least-Squares Collocation (LSC). The final estimation was carried out using Windowed LSC (WLSC). The mean accuracy of the geoid model, when compared against GSVS17 GPS/leveling, is at the 2.4-2.8 cm

level. The second and third models are two of the most widespread global geoid models, namely EIGEN-6C4 (Förste et al., 2014) and EGM2008 (Pavlis et al., 2012), both computed up to degree and order 2190 on the same 2' × 2' grid of ColWLSC2020.

For their comparison, a plane fitting their differences has been estimated and removed to reduce the impact of possible mismatches due to different reference frames and height datums. Table 5 shows the main statistics of the differences between these three geoid models.

**Table 5. Statistics of the differences between the analysed Colorado geoids.**

| Geoid A | Geoid B | Grid Resolution | Std [m] | Min [m] | Max [m] |
|---------|---------|-----------------|---------|---------|---------|
| ColWLSC2020 | EIGEN-6C4 | 2' × 2' | 0.047 | -0.360 | 0.234 |
| ColWLSC2020 | EGM2008 | 2' × 2' | 0.049 | -0.355 | 0.209 |
| EIGEN-6C4 | EGM2008 | 2' ×2' | 0.025 | -0.046 | 0.061 |


From these statistics, it can be noted that over this area the two global models are quite in agreement, with differences of at most few centimeters. When compared to the local model, the amplitude of the differences increases of one order of magnitude with a standard deviation that becomes almost double the standard deviation of the residuals between the two global models.

Figure 14 shows the spatial distribution of such residuals.

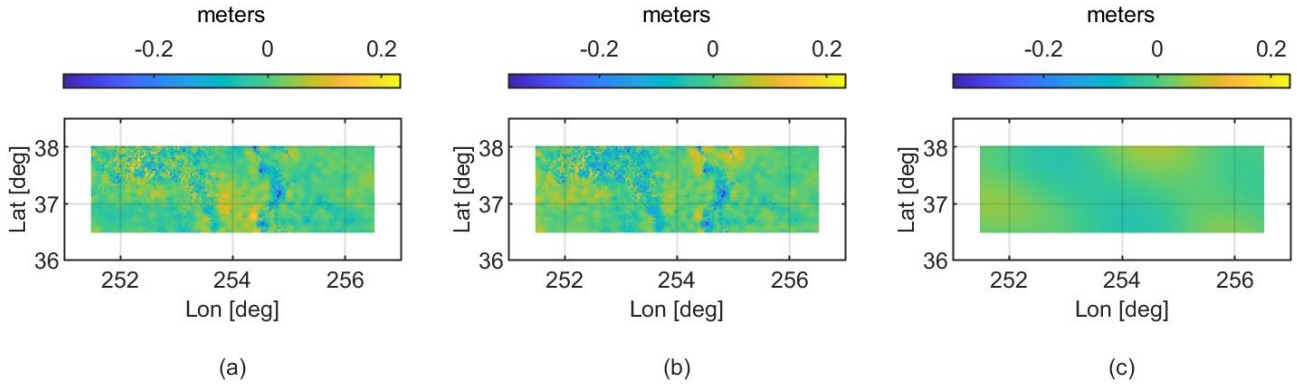

**Figure 14. Difference between the analysed Colorado geoids: (a) ColWLSC2020 – EIGEN-6C4; (b) ColWLSC2020 – EGM2008; (c) EIGEN-6C4 – EGM2008.**


As one can see, the main differences can be found on the Rocky Mountains along the meridian at 255° E and, in smaller part, in the NW region of the considered area. Taking into account that the standard deviations σ of the residuals between the local geoid model and the two global models are almost identical, the agreement between these models can be evaluated on the basis



of the percentage of grid points where all the three models differ by less than one, two, three times σ. As shown in Figure 15, 70% of the points differ by less than 0.049 m, 93% by less than 0.098 m and 98% by less than 0.147 m.

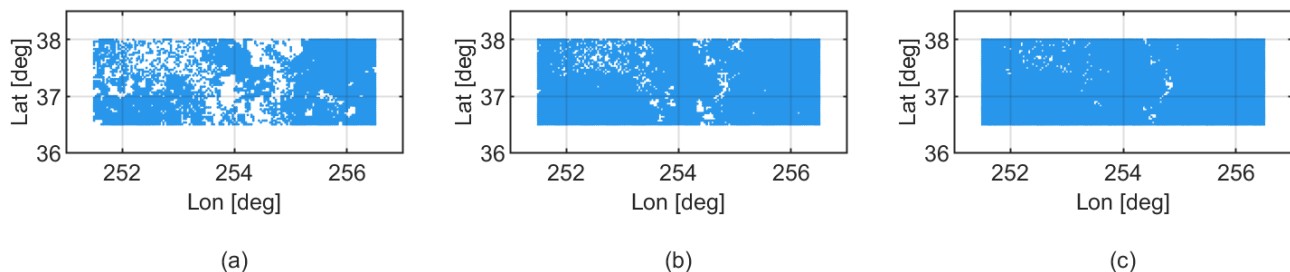

(a)  (b)  (c)

**Figure 15. Agreement of local Colorado geoid model with respect to both global models: (a) residuals below 1σ; (b) residuals below 2σ; (c) residuals below 3σ are shown in blue**

Being the model ColWLSC2020 compared against GSVS17 GPS/levelling with residuals of about 2.5 cm, it seems that the global models suffer of missing information in this area, making the local geoid model more reliable for investigations requiring high level of accuracy.

## 5 Conclusions

ISG is an official worldwide IAG service hosted at Politecnico di Milano aiming at supporting the geodetic community. One of the main activities of ISG is to collect and redistribute regional geoid models at a worldwide scale, making available to users geoid models published by different authors, at different resolutions, referring to different editions and areas, from local to global scale. Before dissemination, the geoids collected in the ISG repository are harmonized through a pre-processing step that validate and convert in a unique format all available data. Since its foundation in 1992, ISG promotes the education for geoid computation, through international schools and special trainings on geoid estimation. The availability of one worldwide archive of regional and local geoids allows to easily find, access and exploit these models, but also to perform comparative analyses, in a research perspective. For instance, one can compare the evolution over time of the same geoid model, as showed in this paper with the assessment of the Japanese geoids of the GSIGEO series. Another example has been provided assessing the three different geoid models available for Sudan and South-Sudan, highlighting the areas of agreement. In addition, the comparison of different geoid models covering the same area have been presented, analysing the area of São Paulo in Brazil covered by both local, national and continental geoids. ISG service is continuously enriching its geoid repository in order to offer to the users a more complete panorama of the geoid models available worldwide and organizing the next geoid schools to support the geodetic community.



**Data Availablity**

All geoid models used in this analysis are freely accessible via the ISG geoid repository where they can be discovered via an alphabetical list or a map (https://www.isgeoid.polimi.it/Geoid/geoid_rep.html). All the DOI-referenced regional geoid models of ISG can be also accessed via https://dataservices.gfz-potsdam.de/portal/?fq=subject:isg.

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
