# Peer review of "Open access to regional geoid models: the International Service for the Geoid"

_Earth System Science Data, 2020_

## Referee Comment (RC1) · Anonymous Referee #1 · 18 Jan 2021

The paper describes the activities of the International Service for the geoid (ISG). This is not a scientific paper in the strict sense, but a description of data and services. Therefore, the ESSD is the appropriate platform for the publication. The language is very good and the paper is easy to follow.

The main activity of ISG is the collection of regional geoid models (usually in gridded format) and providing a download service. Global models (as spherical harmonics) are not stored at ISG, but at ICGEM. ISG is a useful independent service. ISG has today a collection of 226 models. This is already impressive, but is far from being complete. However, this is not the fault of IGS, since they are dependent of the willingness of people providing their models to the service. Not all stored models are freely available. Except of the collection of geoid models, a rather important activity of ISG is the organization of geoid schools. They are organized every several years in many regions of the world. This is very useful for education and spreading the knowledge of geoid determination (and the gravity field in general). The recently established indexing and DOI service as well is very helpful for many authors of geoid models. In my opinion, the height conversion tool is a nice tool and it works, but perhaps not as useful for many people, since most users will prefer to download the whole model and do the calculations themselves. So, they could use their own interpolation methods, not provided by ISG. A useful extension of this service would be, if someone could as well use the restricted models for online transformations (perhaps only for a limited amount of points, to avoid that the users can easily re-generate the original model). The other services provided by ISG (publications and software) are only mentioned shortly in the paper and not treated in detail. The software provided is rather limited and the Newton's Bulletin seems to be inactive since several years.

A big part of the paper (chapter 4) deals with the comparison of geoid models. It shows some of the possibilities of the data of ISG in different application cases. This part, in my opinion, is a little bit long, but I have no objection to leave it as it is. Chapter 4.4 describes the comparison of local with global models. Of course, you can and should do this. But this is not possible with data stored at ISG only. You have to get the global models from elsewhere. It would be nice, if the users could perform such comparisons online. However, I know that this would be a rather big extension of the ISG website.

On line 70 is mentioned "The paper closes with some general considerations and future perspectives of the service". But there is not much there, except of the very last sentence. I would like to learn a little bit more about the future of the ISG (organisation, extension of website, publications, software, ...). This is the main reason why I put in the evaluation "minor revisions".

---

## Referee Comment (RC2) · Anonymous Referee #1 · 18 Jan 2021

line 153: you should explain somewhere in a few words what a geometric and hybrid geoid model is. line 154: besides of geoid and quasigeoid, you have as well simple "transformation surfaces", which serve for a simple realisation of any height system. line 164: This sentence is not complete. Figure 3: This figure is very busy. Is there a way to simplify it? line 377: instead of "will be carried out", write "was carried out".
* * *

---

## Referee Comment (RC3) · Anonymous Referee #2 · 25 Jan 2021

[referee-annotated manuscript omitted]

---

## Referee Comment (RC4) · Anonymous Referee #1 · 11 Feb 2021

Dear Authors, Thank you for your explanations. If you made these changes, I would be satisfied with the revisions. But is there a way I can see the new version?

---

## Author Response (AR1)

Dear Editor,

We would like to thank the Reviewers for their valuable comments and suggestions. We really appreciated their positive general assessment of the submitted paper. Since some of the remarks of the two Reviewers are similar, in the following we will jointly reply to them to better clarify the modifications implemented into the revised manuscript.

As for the height conversion tool, it is currently under monitoring to check how often it is used and which setups are the most requested by the users. We totally agree with the Reviewers that this kind of processing can be performed externally by any arbitrary interpolation, especially by the scientific community. However, this service was mainly thought for technicians and commercial users, often requiring a quick height conversion. Of course, other interpolation methods can be included into the service. Note that the input geoid models are gridded, therefore the bilinear interpolation is actually a spline interpolation. Regarding the use of on-demand and private geoid models as input for the conversion tool, at the moment we do not have the rights to use them for policy reasons.

As for the software and Newton's Bulletin sections, the motivation why they are only mentioned in the paper is twofold: firstly, these services are quite poor, as the Reviewer #1 correctly noticed; secondly, the focus of the paper is on the geoid repository and its exploitation.

As for the format description of the geoid files stored in the ISG repository, we understood the point of Reviewer #2, but we believe that this description is an important piece of information of the paper, also considering the target of the ESSD journal. Moreover, the choice of the unique format for all the models is a crucial step forward a better interoperability and exchange of data. This is something that we would like to emphasize in this work.

As for section 4, the fact that it is quite lengthy has been highlighted by both the Reviewers. Therefore, it has been reduced by removing the comparison of overlapping geoids. In particular, we removed the paragraph concerning the South American area. Moreover, Reviewer #2 asked for clarifications regarding the way in which we managed different reference frames and epochs. As stated in the paper, geoid models were preprocessed by removing a linear trend on the residuals of their differences, thus considering possible systematic differences like the ones due to different reference frames and height datums.

As for the difference between geoid and quasi-geoid, as pointed out by the Reviewer #1, we added a sentence in the introduction to clarify their definition and how we used the two terms throughout the paper.

As for the fact that Figure 3 is quite busy, as again pointed out by the Reviewer #1, this is true, but we believe that this is due to the complexity of the data flow between ISG and GFZ Data Services. In order to improve the figure readability, we added an explanation of the meaning of the arrows in the caption.

Finally, as for the corrections suggested by Reviewer #2 in the annotated manuscript, they were seriously considered by modifying the text accordingly, please see the attached manuscript file with the correction tracking.

---

## Author Response (AR2)

Dear Editor,

First of all, we would like to thank you very much for your revision and the useful remarks and corrections. We seriously took all of them into account and we applied almost all the suggested changes to the manuscript, apart from the removal of Subsections 3.1.1, 3.2.1 and 3.2.2. We would like to keep the current structure of sections and subsections to better separate the subjects. Note that we completely removed Section 4.2, as required. Finally, we left the sentence that there is (generally speaking) some missing information in the global models in Colorado, because we do not know the actual gravity data they used in the area and we prefer to avoid speculations stating that there is a limitation of terrestrial data. On the other hand, we clarified that the two used global models are based on the same terrestrial gravity data set.